# Cryo EM structures map a post vaccination polyclonal antibody response to canine parvovirus

Samantha R. Hartmann [1,5], Andrew J. Charnesky[1,5], Simon P. Früh[2], Robert A. López-Astacio[2], Wendy S. Weichert[2], Nadia DiNunno[1], Sung Hung Cho[3], Carol M. Bator[3], Colin R. Parrish [2] & Susan L. Hafenstein [1,3,4✉]

Canine parvovirus (CPV) is an important pathogen that emerged by cross-species transmission to cause severe disease in dogs. To understand the host immune response to vaccination, sera from dogs immunized with parvovirus are obtained, the polyclonal antibodies are purified and used to solve the high resolution cryo EM structures of the polyclonal Fab-virus complexes. We use a custom software, Icosahedral Subparticle Extraction and Correlated Classification (ISECC) to perform subparticle analysis and reconstruct polyclonal Fab-virus complexes from two different dogs eight and twelve weeks post vaccination. In the resulting polyclonal Fab-virus complexes there are a total of five distinct Fabs identified. In both cases, any of the five antibodies identified would interfere with receptor binding. This polyclonal mapping approach identifies a specific, limited immune response to the live vaccine virus and allows us to investigate the binding of multiple different antibodies or ligands to virus capsids.

[1] Department of Biochemistry and Molecular Biology, The Pennsylvania State University, University Park, PA 16802, USA. [2] Baker Institute for Animal Health, Department of Microbiology and Immunology, College of Veterinary Medicine, Cornell University, Ithaca, NY 14853, USA. [3] Huck Institutes of the Life Sciences, The Pennsylvania State University, University Park, PA 16802, USA. [4] Department of Medicine, The Pennsylvania State University College of Medicine, Hershey, PA 17033, USA. [5] These authors contributed equally: Samantha R. Hartmann, Andrew J. Charnesky. ✉email: shafenstein@psu.edu

Canine parvovirus (CPV) emerged in the mid-1970s as a variant of a virus closely related to feline panleukopenia virus (FPV). The ancestor of CPV gained the canine host range by mutation, and subsequently caused a pandemic of disease in dogs during 1978[1,2]. Multiple variants have emerged and CPV has continued to spread widely[1,2]. Extensive genetic and biochemical studies have shown that specific mutations displayed on or near the capsid surface alter the CPV recognition of the canine transferrin receptor type-1 (TfR) binding site[3–7]. Since the specific host ranges of canine and feline parvoviruses are primarily controlled by the ability of the capsid to bind the TfR, mutations in the receptor binding site alter the ability of the virus to infect different hosts[3,8,9]. Importantly, the site of TfR binding overlaps on the capsid surface with known antigenic epitopes, so that many mutations alter both antibody recognition and host range[10–16].

The small and robust parvovirus has a 26-nm diameter, T = 1 icosahedral capsid that packages a single-stranded DNA (ssDNA) genome of about 5000 bases. The capsid shell is composed of VP2(~90%) and VP1 (~10%), which are generated by differential messenger RNA splicing events such that the sequence of VP2 is also contained within VP1. The structural proteins fold into eight-stranded, anti- parallel β-barrels which are connected by loops. These connecting loops make up most of the capsid topology including a raised region known as the threefold spike that surrounds each icosahedral threefold axis and is a known antigenic site.

Typically CPV is a potent antigen that gives rise to high levels of antibody after infection[17–19]. In addition, the live virus immunization is provided to puppies followed by a series of booster shots[10,20]. In dogs and cats, the antibodies that are produced against the capsids of CPV and FPV efficiently protect against infection in recovered or vaccinated animals. Antibodies against CPV also efficiently cross-neutralize other related viruses such as FPV and mink enteritis virus[21–23].

Previously, the epitopes on parvoviruses have been examined using a variety of approaches, and mostly using panels of rodent monoclonal antibodies (mAb) that were generated against purified capsids injected in the presence of Freund's adjuvant[22,24–29]. Two major binding regions on the CPV and FPV capsids, the A and B site, were identified based on competitions between antibodies for binding, and also from the patterns and locations of mutations in the surface of the capsid that affected antibody binding[22,24]. Cryogenic electron microscopy (cryo EM) analysis of the Fragment antigen-binding (Fab) of eight different mouse and rat mAb complexed with capsids revealed that the binding sites of the antibodies could also be divided into clusters surrounding the A and B epitopes. Each epitope had multiple Fab footprints overlapping with each other and the known antigenic

site. Collectively, the Fab footprints covered about 70% of the exposed surface of the capsid[30]. In addition, the area of overlap shared by each group coincided with the location of residues that were naturally variable in some CPV or FPV isolates[31]. More recent studies have determined the higher resolution structures of two of those antibodies in complex with the capsid, so that the atomic contacts are now known[11,32]. However, it is not known how the rodent mAb resemble canine antibodies produced after infection by a wild virus or vaccination, which result in much larger and prolonged antigen exposure to tissues[33–37].

Recent advances in cryo EM data collection and analysis have made it possible to achieve atomic and near-atomic resolution structures. The software and hardware technical advances have also made it possible to advance innovative mapping of polyclonal responses[12–14,38]. Subparticle reconstruction approaches can be used to identify and classify asymmetric structures, including different antibodies binding to the same or overlapping sites on icosahedral viruses[4,32]. Here we use a subparticle approach to map the binding of the polyclonal IgG responses by visualizing each Fab bound directly onto the icosahedral capsid[15,16,39,40]. We collected the canine antibodies produced by two different beagle puppies (8- and 12-weeks old) after infection with a modified live CPV vaccine. The IgGs were digested and Fab purified, the polyclonal Fabs were incubated with CPV capsids, and the structures were solved by cryo EM. The icosahedral maps revealed Fab density at the antigenic A and B sites. In each of the two data sets, subparticle classification and refinement showed only one or two classes of Fab bound per epitope. Thus, we identified a remarkably focused antibody response. Notably, for both dogs, one region of the B-site was specifically targeted by antibody recognition from separate canines, whereas the A site Fabs bound with different angles and footprints. We identify the conformational epitopes and describe the antibody footprints. All the Fab footprints (A- and B- site binding Fab) overlap with the TfR binding site indicating that any one of the antibodies may neutralize by blocking receptor. Here we show a polyclonal response to an icosahedral capsid that has been defined at high resolution and the Fabs mapped to the capsid.

## Results

To define the CPV-specific antibody response we collected and mapped all CPV-antibodies, as Fab bound to the capsid. Two methods were used to prepare the polyclonal Fab for structural analysis. 1) In the first (Fig. 1) the IgG were purified and digested, the Fabs further purified by binding to an affinity column of intact capsids, Fab eluted, and the resulting affinity purified Fab incubated with purified capsids. 2) In the second method (Fig. 2), the IgGs were isolated and digested, all Fabs were collected, and

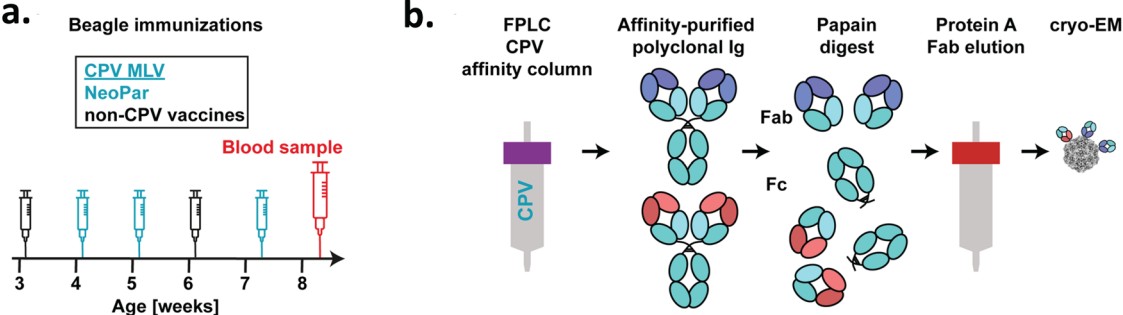

**Fig. 1 Isolation of canine CPV-binding Fab by affinity chromatography. a** Timeline showing CPV modified live virus (MLV) immunizations (with vaccine names), and day of blood sample collection from the male Beagle analyzed in this study. **b** Purification strategy for the isolation of specific CPV-binding Fab by affinity chromatography for cryo EM.

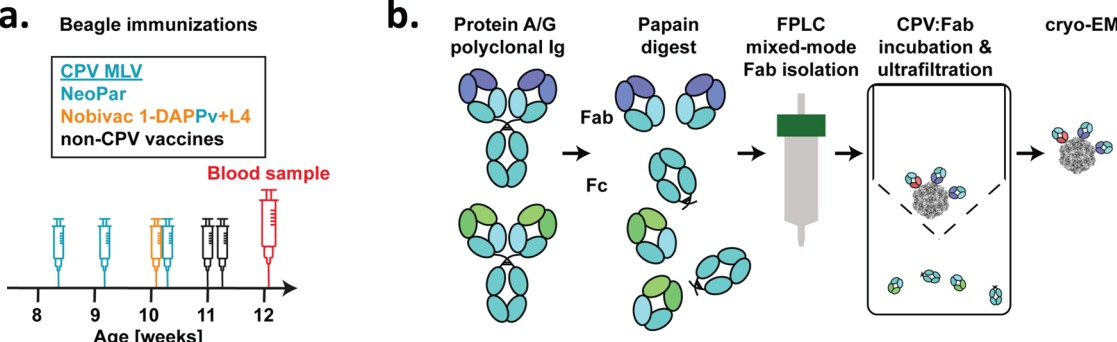

**Fig. 2 Isolation of total-Fab by mixed-mode chromatography. a** Timeline showing CPV modified live virus (MLV) immunizations (with vaccine names) and day of blood sample collection of the Beagle analyzed in this study. **b** Strategy for the preparation of total polyclonal Fab from papain-digested total Ig by mixed-mode chromatography. The total-Fab mixture was incubated with purified CPV capsids, complexes concentrated and washed to remove unbound Fab.

the total Fab repertoire incubated with the capsids, followed by several washes that removed the unbound Fab.

**Isolation and affinity purification of polyclonal CPV-specific Fab.** A blood sample was collected from an 8-week-old male beagle puppy that had been immunized three times with canine parvovirus modified live vaccine (MLV) doses at regular intervals (Fig. 1) (Supplementary Table 1). A high titer of CPV-specific antibodies in the plasma indicated an active response in the puppy to at least one of the vaccine doses (Supplementary Fig. 1). The plasma was run three times on a CPV-affinity column prepared from CPV capsids coupled to an NHS-activated Sepharose column[41]. CPV-specific antibodies were removed each time as indicated by a reduction of the ELISA titer in the plasma before and after each run (Supplementary Fig. 1). Binding of anti-CPV antibodies in the mobile phase (diluted canine plasma) to the stationary phase (CPV capsids coupled to Sepharose) occurred at neutral pH of the plasma, and CPV-binding canine Ig was eluted from the column at pH 3.0. The affinity-purified Ig was digested by papain and the Fab fragments were separated from Fc fragments by Protein A purification (Fig. 1).

**Affinity purified polyclonal Fab-virus complex map reveals bound Fab.** CPV capsids were incubated with the affinity purified polyclonal Fab sample, concentrated, applied to grids, vitrified, and cryo EM data were collected. Particles of approximately 26 nm in diameter could be seen sparsely decorated with Fab in the cryo EM micrographs (Fig. 3). Using a standard processing pipeline, 184,559 particles from 6907 micrographs were extracted for a reconstruction of the affinity purified polyclonal Fab-virus complex (Supplementary Table 2). These particles underwent 2D classification followed by a homogenous refinement where icosahedral symmetry was enforced. In the 2D classes faint ghostlike Fab density could be seen around the capsid (Fig. 3). Similarly, in the central section of the icosahedral map the barely discernable Fab densities illustrated the significantly lower density magnitude of Fab relative to capsid, indicating only a few Fab bound to each capsid. The refinement process concludes with two map outputs: a lower resolution unsharpened map that encompasses all density within the map volume, and a sharpened map that masks out lower magnitude noisy densities to focus on the consistent symmetric densities, which improves the resolution of all icosahedrally ordered densities. The final sharpened, icosahedrally averaged map was refined to 3.2 Å, but the lower density features of asymmetrically bound fab were lost (Supplementary Fig. 2). However, in the unsharpened map, these lower density, asymmetric features were visible and made it possible to begin to

identify where different populations of polyclonal Fab were bound to the capsid (Fig. 3).

**Subparticle extraction and classification reveal discrete A site and B site binding Fab.** To visualize the Fab bound to CPV from the affinity-purified polyclonal preparation we proceeded with a localized reconstruction approach. Subparticles were generated to encompass the previously defined A and B sites of the virus, and the region extending apically from the capsid where antibody fragments (or noise) could be seen. Thus 60 subparticles were made for each binding site on each virus particle, making 11,073,540 A site subparticles and 11,073,540 B site subparticles (Supplementary Table 2). The A and B site subparticles were then classified to sort out those with Fab bound and those without Fab bound. Importantly this classification step would also identify different Fabs that recognize the same epitope but bind in different orientations. Furthermore, because of the size designated for the subparticles, this method would have detected all Fabs that recognized any surface structures, regardless of binding positions or orientations.

For the A site, where definite Fab density was seen in the unsharpened map, the classification of subparticles revealed one single class containing Fab density. The class that had Fab density was then selected and the class was further refined. For the A site 21.4% of the subparticles had Fab density which underwent further refinement to a resolution of 3.8 Å. Although the magnitude of density seen at the B site of the unsharpened icosahedral map was weak, this area was selected for subparticle generation to further explore the density present. The B site subparticle classification also revealed only one class that contained antibody fragment density which was selected and further refined. For the B site 21.4% of the subparticles had Fab density, which was refined to a resolution of 3.7 Å (Fig. 4).

**Total canine Ig purification to produce Fab-virus complex.** In our second approach we used all the polyclonal Fab from the dog serum; forgoing any purification according to specificity (total-Fab) (Fig. 2). A blood sample was collected from a 12-week-old male Beagle that had been immunized four times with canine parvovirus MLV vaccines at regular intervals (Fig. 2) (Supplementary Table 3). As before, a high titer of CPV-specific antibodies detected in the plasma indicated an immune response. Using a protein A/G resin canine Ig were isolated and digested with papain (Fig. 2 and Supplementary Fig. 3). Fab were purified and the total-Fab mixture was incubated with the capsids. Most of the unbound, non-specific Fab were removed by washing the total-Fab-CPV complexes in a 100 kDa cutoff filter to reduce

**Fig. 3 Icosahedral Reconstruction of CPV-affinity purified Fab complexed with CPV. a** Representative micrograph where spherical virus particles can be seen with few Fab decorating the capsids. **b** 2D classification of particles illustrates the lower density of bound Fab (red arrows). **c** Central section of icosahedral reconstruction where capsid can be seen with weak Fab densities (white dashed line) bound to the exterior suggesting that the purified Fab was of relatively low concentration, which was overcome by the subvolume reconstruction approach. **d** The refined sharpened (left) and unsharpened (right) icosahedral reconstruction maps colored radially according to the color key.

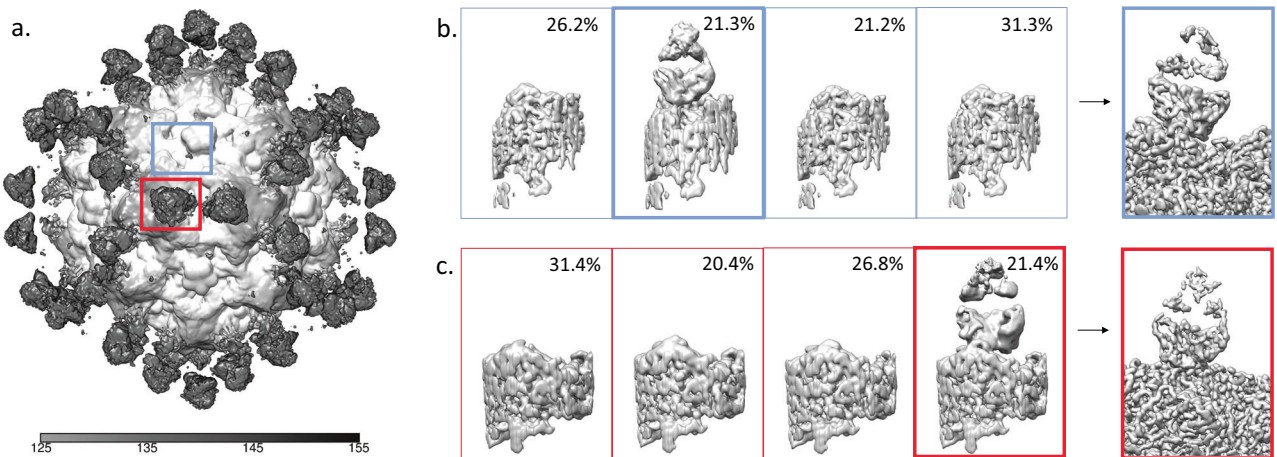

**Fig. 4 Affinity purified Fab Subparticle Classification and Refinement. a** Icosahedrally averaged, unsharpened map showing Fab density over the A and B sites (boxed in red and cornflower blue, respectively). Note the tiny blob of averaged density corresponding to the Fab bound to the B site. **b** 3D Classification of B site subparticles, with one Fab density class outlined in bolded blue that was used for the 3D refinement of the B site Fab to 3.6 Å resolution. **c** 3D Classification of the A site subparticles, with one Fab density classes (bolded red) that was used for the 3D refinement of the A site Fab to 3.7 Å resolution. Based on the classification there are about 12 of each Fab bound per capsid.

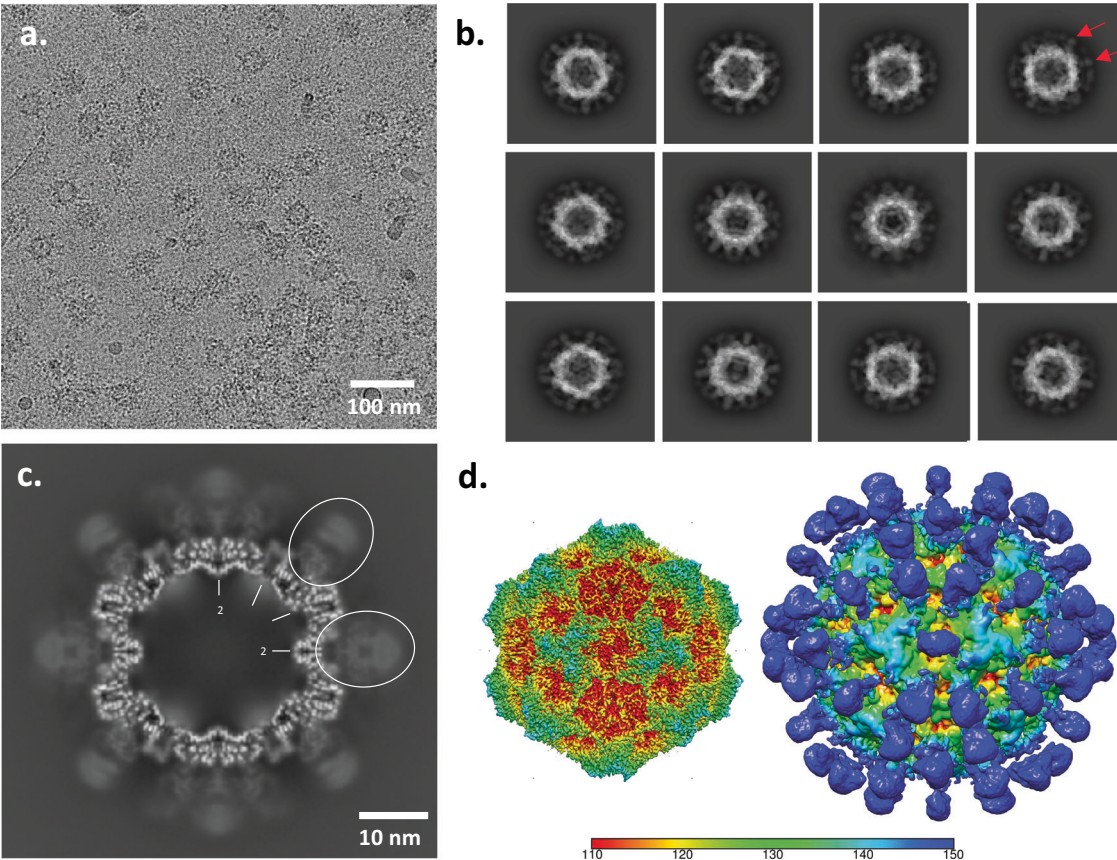

**Fig. 5 Total-Fab incubated with CPV for complex reconstruction. a** Representative micrograph where spherical virus particles can be seen with Fab decorating the capsids. In this incubation, there is more Fab per capsid than in the previous experiment. **b** 2D classification of particles illustrates stronger magnitude of Fab density (red arrows) compared to the affinity purified complexes. **c** Central section of icosahedral reconstruction where capsid can be seen with Fab density (white dashed lines) of slightly lower magnitude bound to the exterior. **d** Sharpened and unsharpened icosahedral reconstructions are colored radially according to the key (left and right respectively).

background in the cryo EM micrographs. The sample was vitrified for cryo EM data collection (Supplementary Fig. 3). (Fig. 2).

**Icosahedral reconstruction of the total-Fab-virus complex clearly shows Fab densities bound to capsid.** Cryo EM micrographs showed virus particles of approximately 26 nm in diameter that were decorated with Fab. Using a similar approach as before for processing the cryo EM data, 322,846 particles from 8,124 micrographs were extracted from the total-Fab purified sample (Supplementary Table 3). 2D classification showed strong Fab density surrounding the capsid (Fig. 5). During homogenous refinement icosahedral symmetry was enforced and the final sharpened map was refined to 3.1 Å (Supplementary Fig. 2). In the unsharpened map, we could see density corresponding to bound polyclonal Fab on the CPV capsid (Fig. 5).

**Subparticle extraction and classification from the total-Fab-CPV complex also revealed discrete A site and B site binding Fab.** Fab like protrusions could be seen on the overall unsharpened icosahedral map of total-Fab-CPV complex. As before, subparticles were generated individually to select for volumes that include the A and B sites of the virus, where antibody fragments could be seen. There were 60 subparticles for each binding site on each virus particle, making 19,370,760 subparticles for each of the epitopes, the A and B site (Supplementary Table 2). The A and B site subparticles were then classified to sort between those with and without Fab bound. This classification identified and

differentiated any overlapping Fab that had different binding configurations.

For the A site, we could see an elongated Fab density in the unsharpened map that rose from the edge of the icosahedral threefold and stretched across the twofold to overlap with symmetry-related density. This density could be interpreted as two different Fab recognizing an A site epitope but with different binding modes. Alternatively, the density might represent a twofold steric clash between symmetry related bound Fab of the same type. Subparticle extraction and classification was used to differentiate between these two possibilities. We found only one class of Fab density corresponding to only one type of Fab recognizing the A site and ruling out the existence of other Fab binding with a different footprint (Fig. 6). This single Fab density represented 9.6% of the subparticles and was refined to a resolution of 3.6 Å.

For the B site, in the unsharpened map there was a bulky Fab shaped density suggesting a possible overlap of different Fab bound similarly to the capsid. After subparticle extraction and classification, two distinct classes of Fab density were identified at the B site (Fig. 6). These two classes were refined separately and the resulting subparticle maps show a different binding location and angle for each Fab. The two refined densities were confirmed as independent Fab at this point and treated separately. The first B site, Fab B1, had 22.5% occupancy and refined to a resolution of 3.0 Å. The second B site, Fab B2, had 22.9% occupancy and refined to a resolution of 3.0 Å, but did contain a loop of poorer density near the interface.

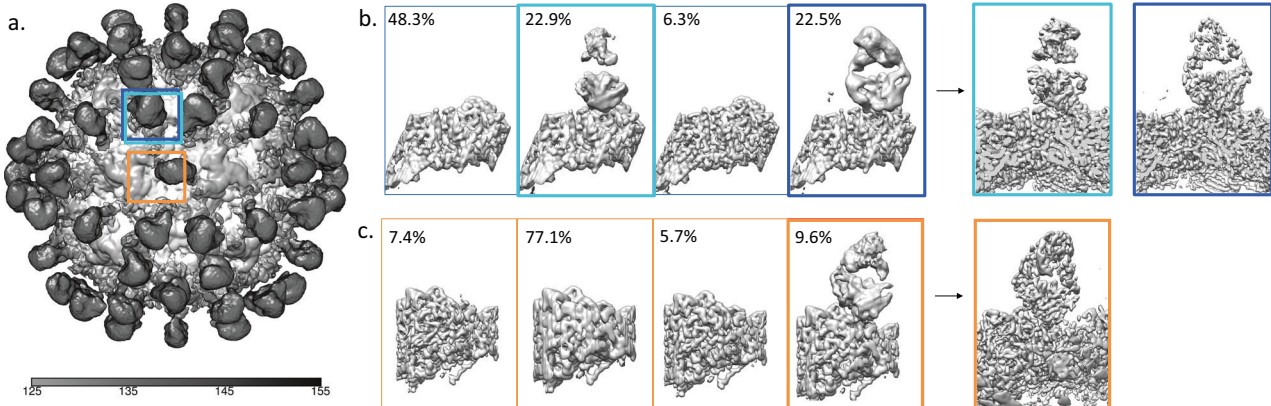

**Fig. 6 Total-Fab-CPV Subparticle Classification and Refinement. a** Unsharpened map of icosahedral reconstruction where Fab density can be seen over the A and B sites (boxed in orange and light blue respectively). **b** 3D Classification of the B site subparticles, with two Fab density classes corresponding to Fab B1 (blue) and Fab B2 (cyan) that were refined independently. **c** 3D Classification of the A site subparticles, with one Fab density class (boxed in orange) that was used for the 3D refinement of the A site Fab. Based on this classification, there are approximately 13 B1 and B2 Fabs per capsid while only about 6A site Fabs per capsid.

**Mapping the Fab footprints**. For each virus-Fab structure, the model building for capsid and Fab was initiated using a virus (PDB ID 2CAS) or Fab (PDB ID 3IY0) crystal structure[30,42]. Since the Fab sequences were not known, alanine was substituted for the residues comprising the variable domain, as has been done previously to interpret the Fab binding[12,13]. In each case the position of the heavy chain complementarity determining region 3 (CDR 3), typically the longest CDR, was able to verify the orientation of light chain and heavy chain. The fab footprints were estimated for both the affinity purified Fab and total-Fab polyclonal samples (Fig. 7, Supplementary Fig. 4). The IgG that recognized the A site differed between the two dogs although the Fab footprints overlap (Fig. 7). Previously defined escape mutations for the rodent IgGs that altered VP2 residues Ser 226 or Thr 228 were found within both A site footprints[24]. The B site binding Fab from the first dog and Fab B1 from the second dog bound with the same angle relative to the capsid producing a similar footprint onto the capsid surface, even though the heavy and light chains were flipped in orientation (Fig. 7). At the interface between Fab B2 and virus, there is a discontinuous density in the light chain CDR1 and all the CDRs are notably shorter for this Fab compared to the others.

**Steric collisions between Fab**. For the affinity purified sample, no symmetry clashes were identified, and 60 copies of the A and B site antibodies could be bound to the capsid. For the total-Fab sample, the A site Fab clashes across the twofold with its own symmetry related neighbor making it possible for only 30A site antibodies to bind to the capsid. The B1 and B2 Fab footprints overlap such that only one or the other can occupy the B site. Furthermore, Fab B2 clashes with the A site Fab. These steric occlusions would reduce occupancy capsid-wide.

**Discussion**

Here, we have identified a polyclonal response from the serum of infected animals by mapping the harvested Fab onto an icosahedral capsid. The traditional structural approach to investigate antibody binding on the surface of an icosahedral virus is to saturate all the potential binding sites with excess Fab generated from mAb. This imposed symmetrical approach often has been successful; however, complexes made with full occupancy of host Fab binding sites on icosahedral viruses do not represent in vivo virus–host interactions. Using a polyclonal Fab sample that was

generated from vaccination allows us to investigate realistic Fab-virus interactions resulting from a mixed population of Fab.

Initially we used an affinity purification approach to select for Ig that recognized the CPV capsid to eliminate any non-specific antibodies. This first approach identified two antibodies, a single Fab bound to the A site and another one bound to the B site. However, the poly Fab concentration was low resulting in weak Fab densities relative to the capsid in the resulting complex. It was possible that the affinity purification step might have skewed results. Thus, we performed another experiment in which we incorporated all the Ig from the second vaccinated dog. The structure that resulted confirmed the initial result showing the one Fab bound to the A site, and two Fab bound to the B site. Thus, there was a similar antibody specificity as seen in the affinity purified sample. Importantly, affinity purification is not necessary and may cause loss of sample. Total poly Fab can be used successfully in polyclonal mapping by cryo EM.

Recently it has been shown that an asymmetric reconstruction and subparticle classification technique can be used to solve a partially saturated Fab-virus complex revealing the structures of Fab- bound and -unbound epitopes[44]. Here, we expand the asymmetric approach[12–14,38] to overcome the heterogeneity stemming from a polyclonal Fab sample, while at the same time dealing with limited binding. In both our experiments the Fab binding sites were not fully occupied, with only about 25 Fab bound per capsid in the first experiment (affinity purified) and 31 Fab per capsid in the second experiment (total Fab). Yet, the subparticle approach was successful in identifying all Fab bound, even when there were steric clashes and overlap of the Fab.

As expected, icosahedral averaging of asymmetrically bound Fabs did not resolve individual Fabs from the polyclonal samples. Importantly, the B site binding Fab from the first experiment (affinity purified Fab) was barely discernable as a small blip of density in the icosahedrally averaged map due to low Fab occupancy (~1 per fivefold) and the local fivefold symmetry averaging. The subparticle reconstruction was necessary to reveal that there was a second antibody bound (Fig. 3).

The results reported here show that the polyclonal antibody responses of dogs to CPV after infection with a live vaccine are surprisingly limited. The analysis clearly revealed only one or two antibodies recognizing each epitope and the existence of only two or three antibodies per dog. Such high specificity has been described before. Polyclonal epitope mapping onto viral proteins and virus capsids has been done previously[12–14,38,40,45]. The

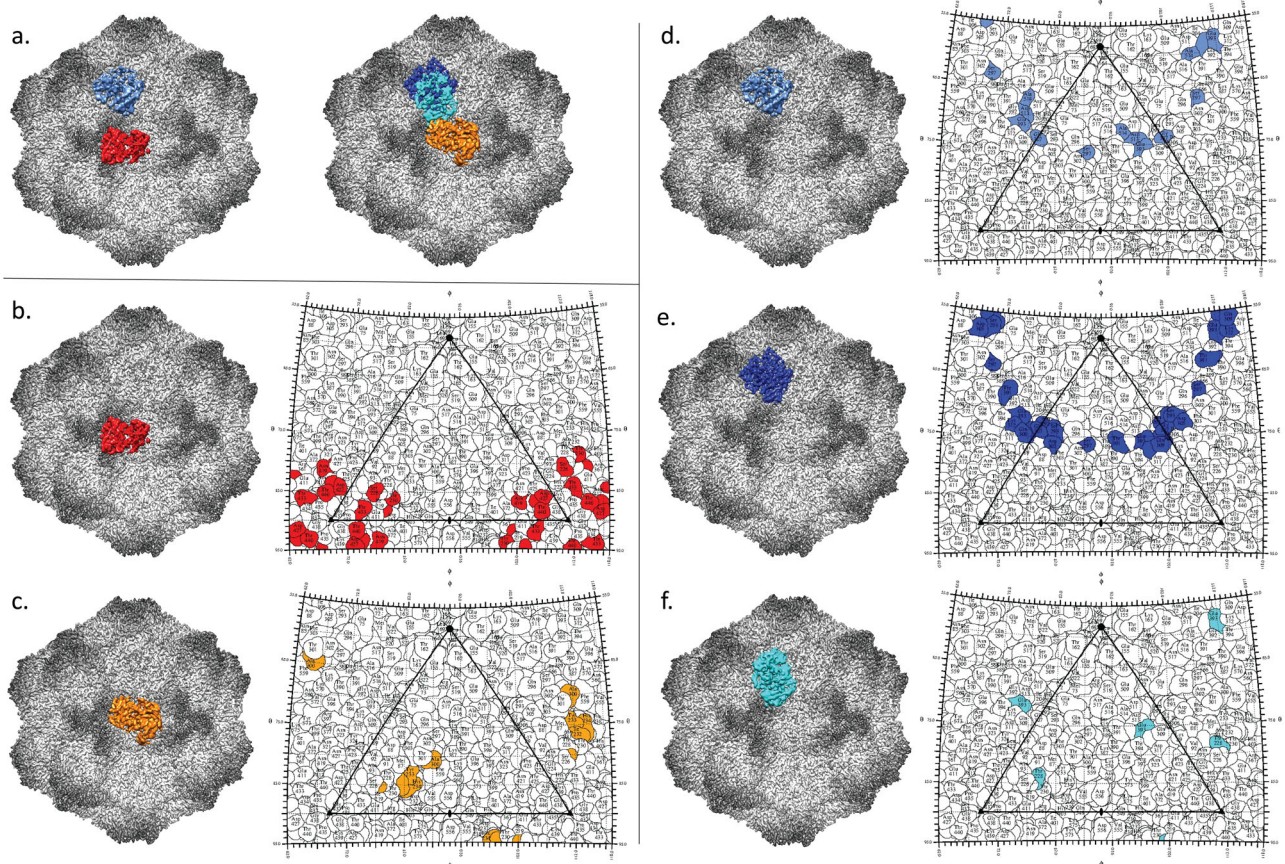

**Fig. 7 Estimated Polyclonal Fab Binding Footprints. a**, left, The two Fab identified from the affinity purified polyclonal are shown bound to capsid (gray scale) with a single Fab recognizing the B site (cornflower blue) and the A site (red) with corresponding roadmap using same color scheme (**b**, **d**). **a**, right, The three Fab identified from the total-Fab polyclonal are shown bound to capsid (gray scale) with A site Fab (orange) and the Fabs B1 and B2 (blue and cyan) with their corresponding footprints shown in roadmaps (**c**, **e**, **f**). Between the two preparations, the A site antibodies have different footprints onto the virus capsid (**b**, **c**). The affinity purified B site footprint is similar to the that of the non-affinity purified B1 footprint, although the heavy chain and light chain have different binding orientations. For the roadmaps the viral surface is shown as a stereographic projection where the polar angles φ and θ represent the latitude and longitude of a point on the viral surface, respectively[43].

number of epitopes that were bound by polyclonal host responses appears to vary during the different stages of the immune response to viruses. For example, the number of epitopes that individuals reacting to influenza H5N1 vaccination recognize on the hemagglutinin (HA) increased from 1–2 (day 1–7 post vaccination) to 3–6 (day 28 post vaccination). In patients surviving an infection with Ebola virus, memory B-cells were sorted, but only two virus reactive mAbs were identified. A cocktail of just these two antibodies was protective in non-human primates against virus infection[45].

Between the two dogs tested, the A site antibodies do share some similarities. However, the Fab recognize the same epitope in different ways, with the Fabs binding the A site from different angles and with different orientations relative to the virus surface. Thus, the A site footprints are different, even though there is some overlap. For the B site, the antibodies generated independently by two different canines bound to the epitope with the same angle and general Fab-capsid interface; however, the Fab seems to be flipped 180°, switching the binding of heavy and light chains of the variable domain. Thus, although the Fab binding footprints are similar what is being recognized by the light chain in the first dog is recognized by the heavy chain in the second dog. This redundant recognition of the B site may be a trend due to the topology of the virus capsid, the exposed region comprising the B site, or a function of canine B-cell recognition. It is also

important to note for the B site, there was one additional Fab that bound differently in the total-Fab sample. This Fab has the shortest complementarity determining regions (CDR) of any Fab in these polyclonal samples. Alternatively, we cannot yet rule out that two dogs responding with highly similar antibodies is coincidence because they use a germline antibody sequence that allows those B cells to be amplified preferentially.

It is probable that the Fab footprints described here underestimate the actual contacts due to the poly alanine Fab structure lacking potentially larger side chains in the variable domain. Likely if we had been able to build from a known sequence, more contact residues would have been identified. Neither A site binding Fab recognized the epitope close enough to the threefold axis to sterically block the symmetry related neighboring Fab, as had been seen previously for the rodent Fab 6 from a previous work (Fig. 3)[30]. Two of the Fabs that recognize the B site, the affinity purified and B1 of the total Fab prep, bind similarly as rodent Fabs 8, 15, and 16 were seen to bind previously (Fig. 3)[30]. The binding of B2 is difficult to compare due to the shorter CDRs, the lack of side chains in the variable domain, and the position of the Fab straddling a cleft in the topology of the capsid (Supplementary Fig. 4).

The relationship between antibody binding sites and antibody escape is complicated by the overlap between the A and B epitopes and the receptor binding site. Since the receptor also

controls the viral host range, virus response to immune pressure may contribute to species jumping. The polyclonal Fab footprints mapped here clearly recognize the known A and B antigenic sites that overlap with the binding site of the TfR[4], which is required for virus infection. The footprints suggest that the mechanism of neutralization is likely blocking subsequent binding of receptor. Notably, any one of the antibodies generated by the dogs could be neutralizing. Thus, what at first appears to be a limited immune response, may actually be redundant. This work identified a specific immune response to vaccination and supports this method for using cryo EM to investigate the complex binding of multiple different antibodies to virus capsids.

Overall, this study confirmed that the immune response to vaccination and the binding of the resulting polyclonal Fab with the CPV capsid is a complex process. Our results identified antibodies that recognized the A and B sites as specific antigenic epitopes. For both the polyclonal Fab-virus preparations prepared, each site had only one or two major antibody specificities. In both cases any one of the identified antibodies would interfere with receptor binding. We also have contributed to the development of methods for using cryo EM to investigate the complex binding of multiple different antibodies or ligands to icosahedral virus capsids that may have a general applicability to different viruses.

## Methods

**Cell and virus preparation**. NLFK cells were grown in 1:1 McCoys 5A-Lebovitz L15 medium (Corning) with 5% FCS in 5% $CO_2$ at 37 °C. Sf9 insect cells were grown in Grace's insect medium (Invitrogen) with 10% FCS andHigh Five insect cells were grown in Express Five serum-free medium (Invitrogen) with 2.4% Glutamine and 10% FCS (Invitrogen) at 28 °C. CPV-2, CPV-2a, CPV-2b and CPV-2 Ala300Asp viruses were produced from infectious plasmid clones in NLFK cells and purified using standard methods by cell lysis, PEG precipitation, chloroform extraction, spinning through sucrose cushion and banding on sucrose gradients to separate empty (light) and full capsids (DNA containing (heavy))[21,22,44,46,47].

**Canine blood samples**. Canine peripheral blood was collected from two male beagle dogs at 8–15 weeks of age in K2 EDTA tubes 7–14 days after the last CPV-containing vaccine dose, and was obtained from Marshall BioResources (North Rose, NY, USA). Details about the immunizations are provided in Supplementary Tables 1 and 2. PBMCs were isolated within one day of blood collection. For separation of cells and plasma/serum, the blood sample was diluted 1:1 in 1x PBS or balanced salt solution and the mixture was layered on top of an equal volume of Ficoll-Paque Density Gradient Media (1.077 g/ml Density Max., GE Healthcare) and centrifugated at $400 \times g$ for 30 min at room temperature. The layer containing the plasma was separated and centrifuged at $1000 - 2000 \times g$ for 10 min at 4 °C to remove platelets. Procedures involving dogs included routine vaccinations that were the standard of care for the client-owned animals, as well as blood collection. Ethical use oversight and all procedures were conducted with the approval of the Cornell University Institutional Animal Care and Use Committee (IAUCUC) (Protocol ID: 2017-0085).

**Preparation of CPV affinity column and affinity purification**. CPV-2b capsids were coupled at 0.481 mg/ml in 0.2 M NaHCO3, 0.5 M NaCl, pH 8.3 to a 1 ml Hi-Trap NHS-activated Sepharose column (Cytiva) for 30 min at room temperature according to the manufacturer's instructions. The column was washed and deactivated and coupling efficiency (47.8%) was determined following

the manufacturer's instructions (PD-10 desalting column method). The column was connected to an Äkta Pure chromatography system, and four empty runs were performed prior to running sample.

CPV-binding canine Ig was isolated by running canine plasma repeatedly on the prepared affinity column for a total of three runs. The column was used at a flow rate of 1 ml/min and sample was loaded at a flow rate of 1–2 ml/min. Running buffer (2 mM $NaH_2PO_4$ with 8 mM $Na_2HPO_4$ and 150 mM NaCl, pH 7.25) was used for equilibration and binding, 0.1 M glycine-HCl, pH 3.0 was used to elute Ig from the affinity column and eluted fractions were immediately neutralized with 1 M Tris-HCl, pH 9.0 and NaCl (150 mM final concentration). Eluted fractions containing Ig were concentrated on 30 kDa MWCO Amicon Centrifugal Filter Units.

**Papain digest of Ig into Fab**. Affinity- or Protein A/G-purified canine Immunoglobulins (Ig) were buffer exchanged in sample buffer (20 mM sodium phosphate, pH 7.0 with 10 mM EDTA) with 7 K MWCO Zeba spin desalting columns (TFS) and digested with immobilized papain (TFS) according to the manufacturer's instructions. Ig at a concentration >0.5 mg/ml were combined with an equal amount of freshly prepared digestion buffer (20 mM sodium phosphate, pH 7.0 with 10 mM EDTA and 20 mM Cysteine-HCl (TFS) and immobilized papain and incubated overnight (10–16 h) at 37 °C. Supernatant containing digested fragment was separated from the resin by centrifugation at $5000 \times g$.

**Affinity purified Fab isolation with Protein A**. Affinity-purified Ig were digested by papain as described above and Fab were separated from Fc and undigested Ig by Protein A purification. Ig were buffer exchange in 0.1 M sodium phosphate with 500 mM NaCl, pH 7.25 with 0.02% Tween-20 and incubated with Protein A resin for 20 min at room temperature with gentle agitation, eluted first with 0.1 M sodium citrate buffer, pH 5.5 followed by elution with 0.1 M glycine-HCl, pH 3.0 and immediately neutralized with 1 M Tris-HCl, pH 9.0. Eluted Fab were buffer exchanged to 50 mM Tris-HCl, pH 7.5 with 150 mM NaCl by repeated dilution and concentration on 10 kDa MWCO Amicon Centrifugal Filter Units (26.7 µg total).

**Fab purification by mixed-mode chromatography**. Canine Immunoglobulins (Ig) were isolated from canine plasma by incubation with Protein A and Protein G resin (TFS) for 80 min at room temperature with gentle agitation, eluted with 0.1 M Glycine, pH 3.0 and immediately neutralized with 1 M Tris-HCl, pH 9.0 and NaCl (150 mM final concentration). Ig were digested by papain as described above. Ig were buffer exchanged in 10 mM sodium phosphate buffer, pH 7.0 by repeated dilution and concentration on 10 kDa MWCO Amicon Centrifugal Filter Units (Millipore Sigma).

Fab were separated from Fc and undigested Ig by chromatography with a mixed-mode support Ceramic Hydroxyapatite Type I column on an Äkta Pure chromatography system (Cytiva). The sample was loaded in 10 mM sodium phosphate buffer, pH 7.0 and eluted with a linear gradient of 40 column volumes from 10 mM to 200 mM sodium phosphate, pH 7.0 (1 ml column volume, 1 ml/min flow rate). Eluted Fab were exchanged in 1x PBS by repeated dilution and centrifugation on 10 kDa MWCO Amicon Centrifugal Filter Units and flash frozen at −80 °C prior to complex formation and cryo EM analysis.

**Western blotting and silver stain**. For western blotting 4.5 µl samples were separated on 10% SDS-PAGE gels and transferred

to nitrocellulose membranes. The membranes were blocked in Tris-buffered saline containing 0.05% Tween 20 and 5% nonfat dry milk and stained with horseradish peroxidase-conjugated rabbit anti-dog Ig (H + L) diluted 1:2500 (Jackson ImmunoResearch) for one hour at room temperature, or overnight at 4 °C. Membranes were washed 3 times in Tris-buffered saline containing 0.05% Tween 20, incubated with Pierce ECL Western Blotting Substrate (TFS) and imaged on a ChemiDoc MP Imaging System (Bio-Rad).For silver stain 4.5 μl samples were separated on 10% SDS-PAGE gels and stained in glass petri dishes with a Pierce silver stain kit (TFS) according to the manufacturer's instructions.

**Cryo EM sample preparation and data collection**. For the affinity purified sample CPV-2a light virus (empty capsid) was added directly to the polyclonal Fab sample resulting in ~1 mL volume and incubated for 10 min at room temperature with periodic mixing by inverting the tube to produce Fab:capsid complexes. Complexes were then concentrated on a 100 kDa MWCO Amicon Ultra-0.5 Centrifugal Filter Unit (nominal pore size 10 nm, recommended ultrafiltration nominal retaining size range 30–90 nm according to the manufacturer [https://www.sigmaaldrich.com/US/en/technical-documents/technical-article/analytical-chemistry/filtration/viral-concentration-amicon-ultrafiltration]) (Millipore Sigma) by spinning at 3000 g for 5 min at room temperature to a total volume of 50 μl and washed twice by dilution with 450 μl 1x PBS and re-concentration to remove non-binding Fab. Small aliquots of complexes in a total volume of 50 μl were directly used for vitrification.

The CPV polyclonal Fab sample was assessed for purity and concentration before vitrification for cryo EM data collection on the Penn State Titan Krios (https://www.huck.psu.edu/core-facilities/cryo-electron-microscopy-facility/instrumentation/fei-titan-krios). 3 μl of the purified virus sample was pipetted onto glow-discharged R2/1 Quantifoil grids (Quantifoil Micro Tools GmbH, Jena, Germany), blotted for 2.5 s, and plunge-frozen in liquid ethane using a Vitrobot Mark IV (Thermo Fisher, USA). Vitrified grids were imaged using a Titan Krios G3 (Thermo Fisher, USA) under automated control of the EPU software. An atlas image was taken at 165× magnification, and suitable areas were selected for imaging on the Falcon 3EC direct electron detector. The microscope was operated at 300 kV with a 70 μm condenser aperture and a 100 μm objective aperture. Magnification was set at 59,000× yielding a calibrated pixel size of 1.1 Å. Four, non-overlapping exposures were acquired per each 2-um-diameter hole of the grid with the beam in parallel mode. The total dose per exposure was set to 45 $e^-/Å^2$. (Supplementary Table 2).

**Icosahedral virus reconstructions**. Icosahedral refinement was performed in cryoSPARC[48]. The micrographs underwent patch motion correction and patch CTF correction. Micrographs were curated and sorted to reject micrographs with crystalline ice. Particles were picked using 2D templates from a subset of particles. Local motion correction was performed on the particle stack and the CTF estimated micrographs. The particles underwent 2D classification. The resulting particles then went into a homogenous refinement. The final resolution was determined by gold standard FSC threshold of 0.143.

**Icosahedral subparticle extraction and correlative classification**. Icosahedral Subparticle Extraction and Correlative Classification (ISECC)[32,49] is a Python- based subparticle extraction package inspired by localized reconstruction[50,51] and block-based reconstruction[52]. A metadata file containing icosahedrally-refined particle origins and orientations is required as input. The beta

version of ISECC was used with cryoSPARC metadata files[48,49]. ISECC_subparticle_extract was used after icosahedral refinement to divide each particle image into subparticles[49]. Both Fab-bound environments were selected for subparticle generation using fullexpand to focus on the Fab bound to the A and B sites. An initial model was made for each subparticle type using relion_reconstruct with 10,000 subparticles. The subparticles were then 3D classified and refined in RELION v3.1[53]. DeepEMhancer was used to improve local sharpening of maps during post-processing[54].

**Atomic model refinement and visualization of capsid and Fab interactions**. Coot[55] was used to substitute the residues in the variable domain of the Fab 14 structure (PDB ID 3IY0)[30] to alanines, as has been done previously[12,13] to initiate the build that was subsequently used for refinement. The structure of CPV (PDB ID code 2CAS)[42] was used to initiate the build of the virus in the complex. The resulting structure was refined in ISOLDE[56] and PHENIX using real space refinement[57]. Validation was by MolProbity[58] bundled in PHENIX[57] (Supplementary Tables 4 and 5). A visual example of the build is provided in Supplementary Fig. 5. Map visualization and images were generated in Chimera and ChimeraX[59,60]. The capsid roadmap was generated by using RIVEM[43].

**Reporting summary**. Further information on research design is available in the Nature Portfolio Reporting Summary linked to this article.

## Data availability

The cryo EM maps and protein structures of the refined polyclonal fab CPV complexes, are deposited in the EM data bank (www.emdatabank.org/) and in the PDB (https://www.rcsb.org). For the affinity purified dataset the A site subparticle with Fab bound (red colored Fab in Fig. 7) (PDB: 7UTP, EMD-26786) and the B site subparticle with Fab bound (cornflower blue colored Fab in Fig. 7) (PDB: 7UTR, EMD-26787) were deposited. For the total-Fab dataset the A site subparticle with Fab bound (orange colored Fab in Fig. 7) (PDB: 7UTS, EMD-26788), the B site subparticle Fab B1 with Fab7UTR, EMD-26787) were deposited. For the total-Fab dataset the A site subparticle with Fab bound (blue colored Fab in Fig. 7) (PDB: 7UTU, EMD-26789), and the B site subparticle Fab B2 with Fab bound (cyan colored Fab in Fig. 7) (PDB: 7UTV, EMD-26790) were deposited. Note: the corresponding whole particle icosahedrally averaged sharpened and unsharpened maps for the respective datasets are available as additional maps under each corresponding accession ID. Source data for the graphs found in the Supplementary materials is available in Supplementary Data 1 and any remaining information can be obtained from the corresponding author upon reasonable request.

## Code availability

Our custom software Icosahedral Subparticle Extraction and Correlative Classification (ISECC) is available on Github https://github.com/goetschius/isecc.

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

## Acknowledgements

Funding was provided by the Pennsylvania Department of Health Commonwealth Universal Research Enhancement (CURE) funds. Research reported in this publication was supported by the Office of the Director, NIH, under award number S10OD026822-01 (S.L.H.), as well as NIH R01AI092571 (C.R.P.). The authors would like to acknowledge the Huck Microscopy Core Facility for use of the Titan Krios and Talos Arctica.

## Author contributions

S.R.H., A.J.C., C.R.P., and S.L.H. designed research; S.R.H, A.J.C., and S.P.F. performed research; S.R.H., A.J.C., S.P.F., R.A.L., C.R.P., and S.L.H. analyzed data; S.R.H., A.J.C., S.P.F., C.R.P., and S.L.H. wrote the paper; N.D., S.P.F., and W.S.W. prepared sample, S.P.F. and W.S.W. collected data. C.M.B., and S.H.C., prepared and collected cryo EM data.

## Competing interests

The authors declare no competing interests.
