## [Peer Review File · Communications Biology]

Reviewers' comments:

Reviewer #1 (Remarks to the Author):

The authors performed an exciting study in which they applied cryo-EM methods to polyclonal antibody:antigen complexes. Their conclusions are supported by the methods and results. Furthermore, the authors successfully utilize methods that have been described for similar biological samples. The manuscript can be improved by addressing the following points which serve to improve clarity, provide a template for reproducibility, and standardize model quality.

1) Page 5, lines 107-108. The authors claim "for the first time, a polyclonal response to an icosahedral capsid has been defined at high resolution and all the component Fab mapped to the capsid unambiguously." While some of the qualifiers (high resolution; all; unambiguously) might make this a correct statement, a quick literature search brought up this citation from late 2011/early 2012: <https://www.ncbi.nlm.nih.gov/pmc/articles/PMC3261347/> . Clearly, not high resolution but I found it remarkable that a similar approach was taken a decade ago with interpretable results, considering it probably pre-dated the direct electron detector. Perhaps this citation should be included with a brief mention in the introduction.

2) Similarly, a recent (Jan 2022) preprint describes this approach for enterovirus: <https://www.biorxiv.org/content/10.1101/2022.01.31.478566v1> . In this example, the entire virion was complexed with polyclonal antibodies, and reconstructions of the pAbs are near atomic resolution. Another citation that should be added if the journal allows preprint citations.

3) Page 18, lines 309-311: this reference is a bit outdated, and if supplanted with the more recent (2021 Nature Communications) <https://pubmed.ncbi.nlm.nih.gov/34376662/>, the approaches are overlapping. The expansion of cryoEM analysis of polyclonal responses was a major aspect of the 2021 manuscript. When considering the citations in points #2 and #3, the authors should take care when claiming novelty of their methods.

4) The introduction is a bit long and reads like an extended abstract. While informative, I feel that it does too well summarizing the entire paper, and takes some of the impact away from Results and Discussion.

5) The discussion could benefit with an explanation or ideas as to why the second approach (isolation of polyclonal IgG, digestion, complex) appears to result in more mapped specificities than using immobilized capsid as a probe to bind IgG from serum. Highlighting this will have a positive impact on the community, so that they choose protocols that lead to the highest diversity of detectable polyclonal responses.

6) I am a bit concerned with the high amount of side chain outliers. Molprobity recommends 0.2% or fewer. Validation report puts these structures in the 25th percentile or so relative to other EM-derived models. Furthermore, the PDB Validation Report values are not always consistent with Supplementary tables. For example, Supp Table 4, B site lists 1.88% Rama outliers, while PDB validation report has zero. Same with Sup table 5, B site 2 (1.21% outliers, but none in the validation report). Please update the tables to reflect the final models that will be accessible from the PDB/EMDB.

7) A few entries also have a large amount of Ramachandran outliers relative to other structures in the PDB. Some have planarity outliers and high clash overlaps (> 0.8). To increase utility of the models in the community, I suggest that these geometry outliers are corrected so that all categories in the validation report summary are in the 75th-100th percentile regions.

A few very minor typos:

Page 3, line 53: drop the "a" in "a eight-stranded, anti-parallel B-barrels"

Page 4, line 79: "Freund's adjuvant" instead of "Fruend's"

Overall, I enjoyed reading this manuscript and found the epitope mapping convincing. Perhaps the biggest highlight was the enrichment of total detectable responses using the protocol B vs protocol A for the IgG isolation and complexing. With a little bit of editing, I hope that other readers will feel the same and grasp the main points. Models with improved geometries will allow the community to use them directly to formulate new hypotheses.

Reviewer #2 (Remarks to the Author):

The manuscript entitled "Post vaccination polyclonal antibody response mapped by cryo EM" by Hartmann et al, describes the mapping of Fab-virus complexes using CryoEM. The Fabs were obtained from two vaccinated dogs and were purified using two approaches. In the first approach, the Ab were first purified using a CPV-affinity column, treated with papain and the purified fabs were subsequently mixed with purified CPVs. On the second approach the total mixture of polyclonal antibodies was first treated with papain and the purified Fabs were incubated with CPVs. The authors used the home-made software Icosahedral Subparticle Extraction and Correlated Classification (ISECC) to reconstruct Fab-Capsid complexes at resolutions around 3 Å. The structures identified 5 different Fabs recognizing two sites on the capsid (A,B). Both sites are close to the TfR and the antibodies would interfere with binding. Results show that the antibodies generated by the two dogs bind site A differently, while the ones binding to site B in a similar fashion. Figures comparing the differences using the atomic models would be very illustrative and helpful. There have been previous studies of other Parvovirus capsid-antibodies structures that show that they shared common features across the family. The authors should discuss their results with the known structures and expand their discussion. This is a nice study that describes a method to map antibody response against parvoviruses that could be used by other groups. The results significance could be increased by putting their results in context with other similar studies. I have only a few comments, once these are addressed, I think the manuscript would be appropriate for publication.

1. Authors should detail if there were differences in the Fab concentrations between the two methods that may explain some of the differences observed between the reconstructions. The complexes using method one seem less populated with Fab.
2. Authors should show a figure with showing the atomic model of the interaction to assess quality of reconstruction.
3. Although the authors show the Capsid interaction footprint, there are no details about how the Fabs interact with each of the sites. Additional figures showing the atomic details is important.
4. Line 113, page 6. The paragraph is a bit confusing about the differences in the different methods, they should refer to Figures 1 and 4 as guide.
5. A comparison between their mapping with previously published parvovirus capsid-fab structures would be very informative.
6. The authors should justify their claim that this is a new method for using cryo-EM. Why is it new? There have been other similar approaches to map antibody responses using cryo-em.

RE: Communications Biology Manuscript COMMSBIO-22-2100-T entitled: "Post vaccination polyclonal antibody response mapped by cryo EM"

Comments of Editor: We are interested in the possibility of publishing your study in, but would like to consider your response to these concerns in the form of a revised manuscript before we make a final decision on publication. We therefore invite you to revise and resubmit your manuscript, taking into account the points raised. Please highlight all changes in the manuscript text file.

Note: Authors responses are in blue font.

The significant delay in preparing the resubmission was due to graduation (and departure) of the first author, adding another author, rebuilding and refining models, and remaking some of the figures. However, we have carefully and thoroughly addressed all comments by reviewers, which has vastly improved the manuscript.

Comments of Reviewers:

Reviewer #1 (Remarks to the Author):

The authors performed an exciting study in which they applied cryo-EM methods to polyclonal antibody:antigen complexes. Their conclusions are supported by the methods and results. Furthermore, the authors successfully utilize methods that have been described for similar biological samples. The manuscript can be improved by addressing the following points which serve to improve clarity, provide a template for reproducibility, and standardize model quality.

1) Page 5, lines 107-108. The authors claim "for the first time, a polyclonal response to an icosahedral capsid has been defined at high resolution and all the component Fab mapped to the capsid unambiguously." While some of the qualifiers (high resolution; all; unambiguously) might make this a correct statement, a quick literature search brought up this citation from late 2011/early 2012: <https://www.ncbi.nlm.nih.gov/pmc/articles/PMC3261347/>. Clearly, not high resolution but I found it remarkable that a similar approach was taken a decade ago with interpretable results, considering it probably pre-dated the direct electron detector. Perhaps this citation should be included with a brief mention in the introduction.

2) Similarly, a recent (Jan 2022) preprint describes this approach for enterovirus: <https://www.biorxiv.org/content/10.1101/2022.01.31.478566v1>. In this example, the entire virion was complexed with polyclonal antibodies, and reconstructions of the pAbs are near atomic resolution. Another citation that should be added if the journal allows preprint citations.

3) Page 18, lines 309-311: this reference is a bit outdated, and if supplanted with the more recent (2021 Nature Communications) <https://pubmed.ncbi.nlm.nih.gov/34376662/>, the approaches are overlapping. The expansion of cryoEM analysis of polyclonal responses was a major aspect of the 2021 manuscript. When considering the citations in points #2 and #3, the authors should take care when claiming novelty of their methods.

We thank the reviewer for recognizing these embarrassing omissions. These citations have been added into the introduction along with one more (Antanasijevic et al 2022

PMID: 35044813) and the text has been modified appropriately to get rid of the grandiose claims.

4) *The introduction is a bit long and reads like an extended abstract. While informative, I feel that it does too well summarizing the entire paper, and takes some of the impact away from Results and Discussion.*

The introduction has been modified significantly.

5) *The discussion could benefit with an explanation or ideas as to why the second approach (isolation of polyclonal IgG, digestion, complex) appears to result in more mapped specificities than using immobilized capsid as a probe to bind IgG from serum. Highlighting this will have a positive impact on the community, so that they choose protocols that lead to the highest diversity of detectable polyclonal responses.*

Discussion has been modified to compare affinity results (1 A site Fab, 1 B site Fab) versus non-affinity purified results (1 A site, 2 B site Fabs). It is clarified that difference is probably due to lower concentration of the poly Fab preparation due to affinity purification steps, and that importantly, affinity purification is not necessary. See Lines 320-332.

6) *I am a bit concerned with the high amount of side chain outliers. Molprobit recommends 0.2% or fewer. Validation report puts these structures in the 25th percentile or so relative to other EM-derived models. Furthermore, the PDB Validation Report values are not always consistent with Supplementary tables. For example, Supp Table 4, B site lists 1.88% Rama outliers, while PDB validation report has zero. Same with Sup table 5, B site 2 (1.21% outliers, but none in the validation report). Please update the tables to reflect the final models that will be accessible from the PDB/EMDB.*

The reviewer's comment led us to rework all structures. Virus and Fab have been rebuilt and refined. New PDB validation reports are included, and side chain outliers range from 0.00 to 0.43% from Molprobit (Supp Table 4 and 5). For the Fab variable domain, alanine residues were substituted, and the model was used to predict footprints. This change did reduce the extent of the estimated footprints. All roadmaps have been recalculated and the text was modified extensively to describe our approach.

7) *A few entries also have a large amount of Ramachandran outliers relative to other structures in the PDB. Some have planarity outliers and high clash overlaps (> 0.8). To increase utility of the models in the community, I suggest that these geometry outliers are corrected so that all categories in the validation report summary are in the 75th-100th percentile regions.*

The structures were rebuilt and refined for the resubmission. Ramachandran outliers now number between 0.00 to 0.19%. Clash scores have now been reduced to a range of 0.10 to 0.21. The data statistics table (Supp Table 4 and 5) now reflects these improvements.

A few very minor typos:

Page 3, line 53: drop the "a" in "a eight-stranded, anti-parallel B-barrels"

Page 4, line 79: "Freund's adjuvant" instead of "Fruend's"

Thanks to the Reviewer, both typos have been corrected.

Overall, I enjoyed reading this manuscript and found the epitope mapping convincing. Perhaps the biggest highlight was the enrichment of total detectable responses using the protocol B vs protocol A for the IgG isolation and complexing. With a little bit of editing, I hope that other readers will feel the same and grasp the main points. Models with improved geometries will allow the community to use them directly to formulate new hypotheses.

We thank the Reviewer for the comments.

Reviewer #2 (Remarks to the Author):

The manuscript entitled "Post vaccination polyclonal antibody response mapped by cryo EM" by Hartmann et al, describes the mapping of Fab-virus complexes using CryoEM. The Fabs were obtained from two vaccinated dogs and were purified using two approaches. In the first approach, the Ab were first purified using a CPV-affinity column, treated with papain and the purified fabs were subsequently mixed with purified CPVs. On the second approach the total mixture of polyclonal antibodies was first treated with papain and the purified Fabs were incubated with CPVs. The authors used the home-made software Icosahedral Subparticle Extraction and Correlated Classification (ISECC) to reconstruct Fab-Capsid complexes at resolutions around 3 Å. The structures identified 5 different Fabs recognizing two sites on the capsid (A,B). Both sites are close to the TfR and the antibodies would interfere with binding. Results show that the antibodies generated by the two dogs bind site A differently, while the ones binding to site B in a similar fashion. Figures comparing the differences using the atomic models would be very illustrative and helpful. There have been previous studies of other Parvovirus capsid-antibodies structures that show that they shared common features across the family. The authors should discuss their results with the known structures and expand their discussion. This is a nice study that describes a method to map antibody response against parvoviruses that could be used by other groups. The results significance could be increased by putting their results in context with other similar studies. I have only a few comments, once these are addressed, I think the manuscript would be appropriate for publication.

1. Authors should detail if there were differences in the Fab concentrations between the two methods that may explain some of the differences observed between the reconstructions. The complexes using method one seem less populated with Fab.

Text has been added to explain affinity purified CPV-specific Fab preparation was less concentrated than the total Fab preparation. This difference can be verified in the central sections (Figures 2 and 5). Previously we have been able to successfully use sub volume approach to reconstruct as few as seven Fabs per capsid (Goetschius et al 2021, PMID: 34074770). The low Fab binding was successfully overcome here, and measure how many Fab were bound per capsid. We have added that information to figure legend 3 that for the affinity purified there were about 12 of each (A and B site) Fabs bound per capsid. In figure legend 6 and results we added that there are about 13 B1 and 13 B2 bound per capsid, although they

occupy different B sites because of clash. Also added that there were only 6 A site fabs bound per capsid.

2. Authors should show a figure with showing the atomic model of the interaction to assess quality of reconstruction.

3. Although the authors show the Capsid interaction footprint, there are no details about how the Fabs interact with each of the sites. Additional figures showing the atomic details is important.

We have modified and added text to clarify our methods. Thank you for pointing out that details are needed. Both results and Methods sections now describe that we took an approach as has been published (Antanasijevic et al 2021 and 2022 PMIDs: 34376662 & 36712368) to replace residues in the variable domain with alanines and use this model to predict the footprints. Additional description has been added and there is a new supplemental figure (Supplemental Figure 4).

4. Line 113, page 6. The paragraph is a bit confusing about the differences in the different methods, they should refer to Figures 1 and 4 as guide.

The text has been clarified and figures are called to help with additional clarification.

5. A comparison between their mapping with previously published parvovirus capsid-fab structures would be very informative.

We thank the Reviewer for this suggestion and have added text to compare the previous work with rodent Fabs (Hafenstein et al 2009 PMID: 19321620).

6. The authors should justify their claim that this is a new method for using cryo-EM. Why is it new? There have been other similar approaches to map antibody responses using cryo-em.

Text has been modified and citations added to include similar approaches.

REVIEWERS' COMMENTS:

Reviewer #1 (Remarks to the Author):

I thank the authors for addressing all of my comments and suggestions. The revised manuscript and details in the rebuttal letter satisfy my original comments and the manuscript is now ready for acceptance.

Reviewer #2 (Remarks to the Author):

The revised manuscript is much improved, and the authors have addressed all my concerns. I recommend publication.